# ORAI1-Regulated Gene Expression in Breast Cancer Cells: Roles for STIM1 Binding, Calcium Influx and Transcription Factor Translocation

**DOI:** 10.3390/ijms23115867

**Published:** 2022-05-24

**Authors:** Mélanie Robitaille, Shao Ming Chan, Amelia A. Peters, Limin Dai, Choon Leng So, Alice H. L. Bong, Francisco Sadras, Sarah J. Roberts-Thomson, Gregory R. Monteith

**Affiliations:** School of Pharmacy, The University of Queensland, Brisbane, QLD 4102, Australia; shaoming.chan@uq.net.au (S.M.C.); a.peters1@uq.edu.au (A.A.P.); l.dai@uq.edu.au (L.D.); c.so@uq.edu.au (C.L.S.); h.bong@uq.edu.au (A.H.L.B.); f.sadras@uq.edu.au (F.S.); sarahrt@uq.edu.au (S.J.R.-T.); greg@pharmacy.uq.edu.au (G.R.M.)

**Keywords:** ORAI1, calcium signaling, breast cancer

## Abstract

A remodeling of calcium homeostasis, including calcium influx via store-operated calcium entry (SOCE), is a feature of breast cancers. SOCE is critical to maintain calcium balance in the endoplasmic reticulum calcium store and is an important mechanism for calcium signaling in a variety of cell types, including breast cancer cells. The canonical mechanism of SOCE is stromal interacting molecule 1 (STIM1)-mediated activation of ORAI. Elevated ORAI1 expression is a feature of basal breast cancer cells. However, the role of ORAI1 in the regulation of transcription in breast cancer cells of the basal molecular subtype is still unclear. Using CRISPR-Cas9 gene editing, ORAI1 protein expression was disrupted in MDA-MB-231 and MDA-MB-468 basal breast cancer cells. The ORAI1 wild-type and mutants were reintroduced into ORAI1 knockout cells to study the role of ORAI1 in gene transcriptional regulation. In the absence of calcium store depletion, ORAI1 regulated *PTGS2* in MDA-MB-231 cells, and this was dependent on ORAI1 pore function and STIM1 binding. The activation of SOCE by thapsigargin resulted in ORAI1-dependent increases in *IL6* transcription in MDA-MB-468 cells; this was also dependent on ORAI1 pore function and STIM1 binding and was associated with the translocation of NFAT1. Given the upregulation of ORAI1 in basal breast cancer cells, our results provide further evidence that ORAI1 may contribute to cancer progression through regulation of gene expression.

## 1. Introduction

Calcium ions (Ca^2+^) regulate an array of cellular functions that are altered during cancer, such as cell motility, proliferation, apoptosis, and gene transcription. Indeed, Ca^2+^ homeostasis itself is remodeled in a variety of cancers and cancer models (reviewed in [1]). Calcium signals are controlled by a diverse collection of channels, pumps, and exchangers, and many of these specific facilitators of Ca^2+^ movement are significantly overexpressed in some cancer types (reviewed in [2]). Indeed, an elevated expression of the ORAI1 calcium channel is a feature of basal-like breast cancers [3], where its expression is critical for breast tumor cell migration and metastasis [4].

Store-operated Ca^2+^ entry (SOCE) is a ubiquitous and essential pathway that regulates Ca^2+^ homeostasis. This calcium entry mechanism is canonically mediated by ORAI1, a Ca^2+^ channel located on the plasma membrane (PM) and is regulated by the Ca^2+^ content of the endoplasmic reticulum (ER). In response to ER Ca^2+^ store depletion, the ER Ca^2+^ sensor protein, stromal interacting molecule 1 (STIM1), oligomerizes and translocates to the ER-PM junctions, where it interacts with ORAI1 and enables a localized Ca^2+^ influx [5]. Ca^2+^ microdomains near ORAI1 increase gene expression through the activation and nuclear translocation of Ca^2+^-dependent transcription factors, such as c-fos and NFAT in rat basophilic leukemia RBL-2H3 cells [6]. Other nuclear transcription factors, such as CREB [7], NF-kB [8], or STAT5 [9], are also regulated by ORAI1-dependent Ca^2+^ entry in different cell models.

ORAI1 can also be activated distinctly from the canonical Ca^2+^ store depletion-mediated STIM1-dependent pathway, referred to as store-independent Ca^2+^ entry (SICE) [10]. SICE includes the activation of ORAI1 by the secretory pathway Ca^2+^-ATPase 2 pump (SPCA2), which is implicated in the promotion of tumorigenesis in breast cancer [11]. Disruption of the ORAI1 coiled-coil domain through a L273S mutation severely reduced the interaction between ORAI1 and SPCA2 [11]. ORAI1 can also assemble with ORAI3 to form an arachidonic acid-regulated Ca^2+^ (ARC) channel [12]. While ARC activation is dependent on STIM1 in HEK293 cells [13,14], in prostate cancer cells, silencing of the STIM1 expression did not alter the ARC activation [15], suggesting a STIM1-independent mechanism. Non-canonical roles for ORAI1 and STIM1 have been identified in diffuse large B cell lymphoma cells, where silencing of the ORAI1 and STIM1 expression impairs migratory behavior, yet this effect was not observed when the ORAI1-mediated Ca^2+^ influx was inhibited using a pharmacological inhibitor [16]. The relative importance of these different ORAI1 cell activation pathways has not yet been assessed in the regulation of gene expression in breast cancer cells. Using a combined approach of gene disruption and rescue experiments, we sought to identify the genes regulated by ORAI1 in basal breast cancer cells, their dependence on Ca^2+^ influx and STIM1 binding, and their association with the translocation of transcription factors.

## 2. Results

### 2.1. Validation of ORAI1 Knockout and Rescue Wild-Type in Basal Breast Cancer Cell Lines

Studies have assessed the consequences of inhibiting or silencing ORAI1 in breast cancer cells and suggested a role for ORAI1 in the proliferation and invasion of these cells [4,17,18]. However, there has been no comprehensive assessment to identify the genes that are regulated by ORAI1 in breast cancer cell lines. As elevated *ORAI1* expression is a feature of basal breast cancer cells [3], we used basal A MDA-MB-468 and basal B MDA-MB-231 breast cancer cells [19] in these studies, which are two of the most widely used basal breast cancer cell lines. Basal A cell lines (e.g., MDA-MB-468) display epithelial characteristics, while basal B cell lines (e.g., MDA-MB-231) display mesenchymal and stem/progenitor-like characteristics [20]. The ORAI1 protein expression was disrupted in MDA-MB-468 cells using CRISPR-Cas9 gene editing, and a wild-type (WT) rescue cell line was created by overexpression of a CRISPR-Cas9-insensitive ORAI1 via lentiviral transduction (Figure 1A–C). *ORAI1* gene editing efficiency and overexpression in the MDA-MB-468 cells were assessed by immunoblotting (Figure 1A) and validated at the functional level using measurement of SOCE (Figure 1B,C). As expected, ORAI1 KO cells exhibited minimal Ca^2+^ entry following ER store depletion by the sarcoplasmic Ca^2+^ ATPase (SERCA) pump inhibitor CPA, while ORAI1 WT rescue MDA-MB-468 cells showed a similar Ca^2+^ influx compared to parental MDA-MB-468 cells. The same strategy was used to knockout the ORAI1 expression in MDA-MB-231 (Figure 1D–F).

### 2.2. ORAI1 Differentially Regulates Basal Gene Expression in MDA-MB-468 and MDA-MB-231 Cells

As ORAI1 regulates a variety of transcription factors [6,21], the consequences of *ORAI1* gene disruption in MDA-MB-468 (Figure 2A and Appendix A Appendix A) and MDA-MB-231 (Figure 2B) cells on gene transcription regulation were first assessed in the absence of calcium store depletion. A list of genes potentially associated with Ca^2+^ signaling was developed, and the mRNA expression was analyzed by RT-qPCR. A minimal difference in the gene expression levels was observed in the ORAI1 KO cells compared to the ORAI1 WT rescue in MDA-MB-468 cells, although there was a modest change in some genes, such as Interleukin 6 (*IL6*) (Figure 2A). In contrast, a robust increase in prostaglandin-endoperoxide synthase 2 (*PTGS2*) expression was evident in the ORAI1 WT rescue MDA-MB-231 cells compared to the ORAI1 KO cells (Figure 2B).

### 2.3. PTGS2 Gene Expression Regulation by ORAI1 in MDA-MB-231 Is Dependent on STIM1 Activation and Calcium Influx

To assess whether the increased expression of *PTGS2* in ORAI1 WT rescue MDA-MB-231 cells was dependent on ORAI1 function, the ORAI1 expression was rescued using the pore dead mutant E106Q [22] or the STIM1 binding-deficient mutant L273D [23]. The expressions and functions of these two mutants were assessed in the MDA-MB-231 cells. The two mutants were expressed at a similar level to the WT ORAI1 rescue construct (Figure 3A); however, unlike the WT rescue, the mutants did not facilitate Ca^2+^ influx after store depletion by CPA (Figure 3B,C). As with SOCE, *PTGS2* expression was not rescued with either the ORAI1 E106Q or ORAI1 L273D mutants (Figure 3D). These results suggest that regulation of the *PTGS2* expression by ORAI1 is dependent on STIM1-mediated Ca^2+^ influx.

### 2.4. Thapsigargin-Mediated Calcium Store Depletion Increases IL6 mRNA Levels in MDA-MB-468 Cells and Is Dependent on STIM1-Mediated Ca^2+^ Influx

Despite an established role for ORAI1 in the basal calcium influx in MDA-MB-468 cells [24], KO and the rescue of ORAI1 in this breast cancer cell line did not significantly change the expression levels of the assessed Ca^2+^-regulated genes. However, since robust activation of SOCE is induced by ER calcium store depletion [25], we assessed the ability of the Ca^2+^ store depletion in MDA-MB-468 cells to effect gene transcription. To identify an appropriate inducer of Ca^2+^ store depletion to activate transcription, the reversible SERCA inhibitor CPA [26] and the irreversible SERCA inhibitor thapsigargin [27] were compared for their abilities to induce changes in the *IL6* mRNA levels. *IL6* was used because the *PTGS2* mRNA levels were undetectable in MDA-MB-468 (data not shown). The *IL6* mRNA levels were induced by thapsigargin more than by CPA, and the effects were greater at 6 h for both SERCA inhibitors (Figure 4A). Therefore, a 6 h treatment with thapsigargin (1 µM) was selected to explore the features of ORAI1 that were critical to the *IL6* expression induction. To determine the dependency of ORAI1 and SOCE on thapsigargin-induced *IL6* gene expression, the ORAI1 pore dead mutant E106Q and ORAI1 STIM1 binding-deficient L273D mutant were overexpressed in MDA-MB-468 ORAI1 KO cells. As seen with MDA-MB-231, these two mutants were expressed at similar protein levels compared to the WT rescue cells (Figure 4B) but failed to restore SOCE (Figure 4C,D). Thapsigargin induced the *IL6* mRNA levels in MDA-MB-468 ORAI1 KO cells, and this increase in *IL6* was further enhanced by an overexpression of WT ORAI1 (Figure 4E). In contrast, rescue with the ORAI1 pore dead mutant E106Q or ORAI1 STIM1 binding-deficient L273D mutants showed a similar *IL6* level to ORAI1 KO cells (Figure 4E). Due to the ability of thapsigargin to induce *IL6* transcription, even in the absence of ORAI1, we compared cytosolic-free Ca^2+^ changes induced by thapsigargin in the presence or absence of extracellular calcium in MDA-MB-468 cells (Figure 4F,G). Under both conditions, thapsigargin induced an initial increase in cytosolic-free Ca^2+^; however, only in the presence of extracellular Ca^2+^ did a sustained calcium influx occur, and this sustained phase was associated with a greater *IL6* mRNA induction.

### 2.5. Blocking Importin-β Function Is Sufficient to Inhibit ORAI1-Dependent Thapsigargin Induction of IL6

SOCE can induce changes to gene transcription by promoting nuclear translocation of transcription factors, including NFAT1 [6]. To explore whether this mechanism was involved in *IL6* mRNA induction, importazole, a small molecule inhibitor of the nuclear transport receptor importin-β, was used to determine whether *IL6* induction was a consequence of nuclear translocation of transcription factors. Using high content imaging, the efficiency of importazole to inhibit NFAT1-GFP nuclear translocation, in response to store depletion and SOCE activation by thapsigargin, was assessed (Figure 5A–C). As expected [28], NFAT1-GFP translocated to the nucleus in response to thapsigargin treatment (Figure 5A–C). Pre-incubation of the cells with importazole greatly decreased NFAT1-GFP nuclear translocation (Figure 5A–C). Moreover, the inhibition of importin-β was sufficient to significantly reduce ORAI1-dependent thapsigargin induction of *IL6* (Figure 5D) in MDA-MB-468 cells, implicating SOCE-mediated nuclear translocation of the transcription factors in *IL6* mRNA induction.

## 3. Discussion

Several studies show an alteration of ORAI1-mediated Ca^2+^ influx and/or a remodeling of the ORAI1 expression in breast cancers [3,17]. ORAI1 contributes to cell proliferation, cell migration, and apoptotic resistance [29]. Despite ORAI1 being linked to several hallmarks of cancer [30] and the activation of transcription factors [7,8,9], ORAI1 regulation of gene transcription, particularly in the context of breast cancer, is still poorly defined. Using a CRISPR-Cas9 and a rescue approach, this study investigated the role of ORAI1 on gene transcription in basal breast cancer cells.

The MDA-MB-468 and MDA-MB-231 cell lines were selected as representative cell models of basal A and basal B breast cancer subtypes, respectively. Even though complete ORAI1 protein depletion was observed in both cell lines, in the MDA-MB-468 ORAI1 KO cells, SOCE was not completely abolished. Residual SOCE in the MDA-MB-468 cells was inhibited by the expression of ORAI1 pore dead mutant E106Q and STIM1 binding-deficient mutant L273D. These data suggest that in MDA-MB-468 cells, ORAI1 is not the sole Ca^2+^ channel responsible for SOCE. Although ORAI1 is considered the main channel responsible for SOCE, ORAI2 [31], ORAI3 [32,33,34], and TRPC1 [35] have been reported to mediate SOCE in various cell models. ORAI1 forms heteromeric channels with ORAI3 and/or TRPC1 [36,37]; thus, the ORAI1 pore dead mutant E106Q and STIM1 binding-deficient mutant L273D may heterodimerize with ORAI3 and/or TRPC1 and inhibit residual SOCE in MDA-MB-468 ORAI1 KO cells via a dominant negative mutant effect. While the MDA-MB-468 WT rescue cells showed a similar Ca^2+^ influx to parental cells, the MDA-MB-231 WT cells showed a reduced SOCE compared to parental MDA-MB-231 cells. Given the stoichiometric importance of the ORAI1:STIM1 ratio (reviewed in [38]), there is likely an excess of exogenously expressed ORAI1 compared to endogenous STIM1, and this would have impacted ORAI1 activity in MDA-MB-231 cells. Taken together, these results highlight cell line-specific differences in ORAI channel-mediated Ca^2+^ influx. An extensive panel of basal A and basal B cell lines would need to be compared to determine whether the phenotype we observed is a differential feature of the two different basal cell line subtypes.

This study identified ORAI1 as a regulator of *PTGS2* and *IL6* expressions in basal breast cancer. *PTGS2* and *IL6* play important roles in the pathogenesis of breast cancer by promoting tumor growth, invasion, angiogenesis, and apoptosis resistance (reviewed in [39,40]). PTGS2, also known as cyclooxygenase 2 (COX2), is a key enzyme in prostaglandin biosynthesis and is critical for regulating the inflammatory response. ORAI1 also regulates EGF-mediated and lipopolysaccharide-mediated *PTGS2* gene expression in colorectal cancer cell lines [41] and AGS gastric adenocarcinoma cells [42], respectively. To our knowledge, this is the first time that ORAI1 has been shown to regulate *PTGS2* expression in breast cancer cells and in the absence of induced Ca^2+^ store depletion. *IL6* was first discovered as a B cell differentiation factor, which induces the maturation of B cells into antibody-producing cells [43]. It is a pleiotropic cytokine involved in bone homeostasis, cardiovascular protection, metabolism, and inflammation (reviewed in [44]). Inflammation and cancer are closely related [45], and *IL6* is associated with nearly all of the hallmarks of cancer [46]. High *IL6* levels correlate with poor clinical outcomes in esophageal squamous cell carcinoma [47] and invasive breast cancer [48]. *IL6* is highly expressed in basal breast cancer cell lines [49], which are also associated with elevated *ORAI1* [3]. *IL6* silencing strongly suppresses in vitro anchorage-independent colony formation and in vivo tumor formation and growth of basal breast cancer cells [50]. Hence, targeting the ORAI1/STIM1 function in basal breast cancer cells could lead to a decrease in overall inflammatory signals through its effect on *IL6* and *PTGS2* expression.

In ORAI1 KO MDA-MB-468, thapsigargin induced *IL6* expression despite the absence of ORAI1. These results align with previous studies in mice, where sustained Ca^2+^ entry, but not the initial increase in Ca^2+^ induced by thapsigargin, was completely lost in lacrimal [51] and testicular [52] cells harvested from ORAI1 KO mice compared to their wild-type littermates. Thapsigargin irreversibly blocks the ER Ca^2+^ pump SERCA, and, consequently, basal Ca^2+^ lost from the ER is not re-sequestered into the ER, resulting in increased intracellular Ca^2+^ levels. This increase in intracellular Ca^2+^ would be more sustained when SOCE is activated [53]. Our results show that this initial increase in intracellular Ca^2+^ induced by thapsigargin, without SOCE, was sufficient to elevate *IL6* mRNA levels by nearly 20-fold compared to untreated cells. However, *IL6* mRNA expression was increased by more than 50-fold in the presence of a sustained calcium influx in the presence of a functional ORAI1 channel.

In addition to its role in SOCE via activation by STIM1, ORAI1 is also activated via other mechanisms. For example, overexpression of SPCA1 induces an ORAI1-mediated Ca^2+^ influx independently of STIM1 [54]. Moreover, silencing ORAI1, but not STIM1, sensitizes MDA-MB-231 basal breast cancer cells to staurosporine-induced apoptosis [29]. Using the ORAI1 E106Q pore dead mutant and L273D STIM1 binding-deficient mutant, our study demonstrated that ORAI1 regulation of *PTGS2* and *IL6* expression in basal breast cancer is dependent on both ORAI1 pore activity and STIM1 binding.

## 4. Materials and Methods

### 4.1. Plasmids

ORAI1 (gRNA #1: GATCGGCCAGAGTTACTCCG, gRNA #2: CGGCGAAGACGATAAAGATC) gRNAs were inserted into lentiCRISPRv2-hygro (gift from Brett Stringer, Addgene plasmid # 98291). AdTRACK-CMV-hORAI1 (gift from Gregory J. Barritt) was used as a PCR template to amplify ORAI1. ECFP-ORAI1-L273D (gift from Donald L. Gill) was used as a PCR template to amplify ORAI1L273D. ORAI1-myc E106Q MO70 (gift from Anjana Rao, Addgene # 22754) was used as a PCR template to amplify ORAI1E106Q. HA-NFAT1(4-460)-GFP (gift from Anjana Rao, Addgene # 11107) was used as a PCR template to amplify NFAT1(4-460)-GFP. All PCR products were cloned into pcDH-EF1-FHC (gift from Richard Wood, Addgene plasmid # 64874) using a strong Kozak sequence (GCC ACC) in front of the starting ATG. To generate ORAI1 cDNA insensitive to CRISPR-Cas9, silent mutations (changing the DNA base pair but keeping the amino acid sequence identical) were introduced on ORAI1 cDNA PAM sequence sites by PCR-driven overlap extension.

### 4.2. Cell Culture

MDA-MB-231 basal breast cancer cells were obtained from American Type Culture Collection. The MDA-MB-468 human basal breast cancer cell line was obtained from The Brisbane Breast Bank, UQCCR, Brisbane, Australia. Cells were cultured in Dulbecco’s Modified Eagle’s Medium (DMEM; Sigma-Aldrich, St. Louis, MI, USA, D6546) supplemented with 10% FBS (Moregate, Bulimba, QLD, Australia, LOT# 50301112) and 4 mM l-glutamine (complete DMEM; Gibco, Waltham, MA, USA, 25030081) in a 37 °C humidified incubator with 5% CO_2_. Cell line authentication was performed with STR profiling at the QIMR Berghofer Institute, Brisbane. Mycoplasma testing was done biannually at the Translational Research Institute, Brisbane, Australia.

### 4.3. Generation of Cell Lines

Lentiviral particles were produced in HEK293T cells with second-generation packaging plasmids and lipofectamine 2000 (Invitrogen, Waltham, MA, USA, 11668019) transfection. To generate ORAI1 knockout (KO) cell lines, MDA-MB-231 and MDA-MB-468 cells were subsequently transduced with lentiCRISPRV2-ORAI1 gRNA viral particles in the presence of 8 µg/mL of polybrene (MilliporeSigma, Burlington, MA, USA, TR1003G). The viral media was replaced after 48 h, and cells were selected with 400 µg/mL hygromycin B (Gibco, Waltham, MA, USA, 10687010) for 5 days. Cells were plated in a 96-well plate at a density of 0.5 cells/well to isolate single cell colonies. To generate ORAI1 rescue cell lines, ORAI1 KO cells were transduced with either ORAI1, ORAI1E106Q, or ORAI1L273D containing viral particles as described above and selected with 2 µg/mL of puromycin (Sigma-Aldrich, St. Louis, MI, USA, P8833). To generate the NFAT1-GFP MDA-MB-468 cell line, MDA-MB-468 cells were transduced twice with HA-NFAT1(4-460)-GFP containing viral particles as described above, selected with 1 μg/mL puromycin, and fluorescence-activated cell sorted (FACS) using BD FACSAria Fusion (Translational Research Institute, Woolloongabba, Australia) to select for the GFP-expressing population.

### 4.4. Immunoblotting

Cells were lysed in protein lysis buffer (50 mM Tris, 100 mM NaCl, 1% NP-40, 0.5% sodium deoxycholate) supplemented with 1x protease and phosphatase inhibitors (Roche Applied Science, Penzberg, Germany, 11836153001 and 04906845001). Gel electrophoresis was performed using Mini-PROTEAN^®^ TGX Pre-cast Gels, and protein was transferred to a PVDF membrane (Bio-Rad Laboratories, Hong Kong, China, 4568084 and 1704157). Membranes were blocked for 1 h in 5% skim milk in phosphate-buffered saline containing 0.1% Tween-20 (Sigma-Aldrich, St. Louis, MI, USA, P9416) (PBST) before incubating overnight at 4 °C with anti-ORAI1 (Sigma-Aldrich, St. Louis, MI, USA #08264 1:4000) or anti-vinculin (Cell Signaling, Danvers, MA, USA # 13901, 1:1000). Goat anti-rabbit horseradish peroxidase conjugate secondary antibodies (Bio-Rad Laboratories, Hong Kong, China, 1706515) were diluted 1:10,000 in 5% skim milk in PBST and incubated for 1 h at room temperature. Proteins were imaged using the SuperSignal West Dura Extended Duration Chemiluminescent Substrate (Thermo Fisher Scientific, Waltham, MA, USA, 34076) on the Bio-Rad ChemiDoc Imaging System (Bio-Rad Laboratories, Hong Kong, China).

### 4.5. Measurement of Intracellular Calcium

Cells were seeded into black-walled 96-well plates (CellBIND; Corning, Corning, NY, USA, CLS3340) at a density of 20 × 10^3^ cells per well, and Ca^2+^ imaging was performed 24 h later using FLIPR^TETRA^ (Molecular Devices, San Jose, CA, USA). Cyclopiazonic acid (CPA; Sigma-Aldrich, St. Louis, MI, USA, C153P)-induced SOCE was performed as previously described [55]. For thapsigargin (Sigma-Aldrich, St. Louis, MI, USA, T9033)-induced ER store depletion, 1 µM of thapsigargin in either nominal or 1.8 mM CaCl_2_ PSS was added after 10 s and assessed for an additional 600 s. Changes in fluorescence intensity relative to baseline fluorescence over time were expressed as cytosolic Ca^2+^ changes (ΔF/F_0_).

### 4.6. mRNA Isolation and RT-qPCR

Where indicated in figures, MDA-MB-468 cells were treated with thapsigargin or CPA at the stated concentration and time, or pre-incubated for 15 min with 40 µM importazole (IMPZ; Selleck Chemical, Houston, TX, USA, 58446). Total mRNA was isolated from cells and purified using the RNeasy Mini kit (Qiagen, Singapore, 74004) according to the manufacturer’s protocol. RNA was reverse transcribed into cDNA using the Omniscript Reverse Transcription Kit (Qiagen, Singapore, 205113). Synthesized cDNA was amplified using TaqMan Fast Universal PCR Master Mix (Applied Biosystems, Waltham, MA, USA, 4352042), and real-time PCR reactions were performed using the StepOnePlus Real-Time PCR System (Applied Biosystems, Waltham, MA, USA) under universal cycling conditions. Relative gene expression was quantitated using the comparative C*_T_* method (ΔΔC_T_), normalized to *ACTB* (MDA-MB-231) or the mean of *PGK1* and *PPIB* (MDA-MB-468). TaqMan gene expression assays used for this study were: *ACTB* (Hs0160665_g1), *AXL* (Hs0164444_m1), *BCL2* (Hs00608023_m1), *CDH1* (Hs00170423_m1), *CDK1* (Hs00938777_m1), *CLDN1* (Hs00221623_m1), *CLDN4* (Hs00976831_s1), *CTGF* (Hs00170014_m1), *CTNBB1* (Hs00170025_m1), *EGFR* (Hs01076092_m1), *IL6* (Hs00174131_m1), *ITPR3* (Hs01573555_m1), *Jun* (Hs01103582_s1), *MYC* (Hs00153408_m1), *ORAI1* (Hs03046013_m1), *ORAI2* (Hs00259863_m1), *ORAI3* (Hs00743683_s1), *PGK1* (Hs99999906_m1), *PMCA4* (Hs00608066_m1), *PPIB* (Hs00168719_m1), *PTGS2* (Hs00153133_m1), *SERPINE1* (Hs00167155_m1), *STIM2* (Hs00372712_m1), *TMEM66* (Hs00211619_m1), and *WWTR1* (Hs00210007_m1).

### 4.7. High Content Imaging and NFAT Nuclear Translocation

NFAT1-GFP-expressing MDA-MB-468 cells were plated at a density of 6 × 10^3^ cells per well into black-walled 96-well plates. After 48 h, media was removed, and cells were treated with either vehicle control (0.1% DMSO) or importazole (40 µM; IMPZ; Selleck Chemical, Houston, TX, USA, 58446) for 15 min prepared in FluoroBrite^TM^ DMEM media (Thermo Fisher Scientific, Waltham, MA, USA, A1896701) before further incubation with thapsigargin (1 µM) for 1 h. Cell nuclei were stained for 15 min with Hoechst 33342 (2 ng/mL; Invitrogen, Waltham, MA, USA, H3570) prior to fluorescence imaging at 37 °C using an epifluorescence microscope, ImageXpress Micro (Molecular Devices, San Jose, CA, USA) with a 10X objective. A single image per well was captured for NFAT1-GFP and Hoechst 33342. For nuclei identification, nuclei were first segmented using the Multi-Dimensional Motion Analysis module in MetaXpress (Molecular Devices, San Jose, CA, USA). Using a collection of previously described image analysis techniques, nuclei were binarized and dilated, border objects were removed, and the nuclei were eroded back to their initial size producing a processed nuclei mask without cells near the border of the image [56]. This processed nuclei mask was combined with the original Hoechst image using the logical AND operation, creating an image that consists of only the Hoechst fluorescence that overlaps with the processed nuclei. NFAT1-GFP translocation to the nucleus was assessed using the translocation module on the MetaXpress software on this image and the original GFP image. Cells with a nucleus-cytoplasm Pearson’s correlation coefficient of more than 0.6 were classified as having positive translocation.

### 4.8. Statistical Analysis

Values represent means ± SD for at least three independent experiments as indicated in the figure legends. Statistical analyses were performed using GraphPad Prism v8.00. Student *t* tests, a one-way ANOVA with Dunnett’s post-hoc and a two-way ANOVA with Šidák’s post-hoc were used as indicated in the corresponding figure legends. The statistical significance was set at *p* < 0.05.

## 5. Conclusions

ORAI1 regulated the expression of *PTGS2* in unstimulated MDA-MB-231 cells. ER Ca^2+^ store depletion increased *IL6* mRNA in MDA-MB-468 cells, and this was associated with transcription factor nuclear translocation. In all cases assessed, ORAI1 regulation of transcriptional activity was dependent on Ca^2+^ transport and activation by STIM1.

## Figures and Tables

**Figure 1 ijms-23-05867-f001:**
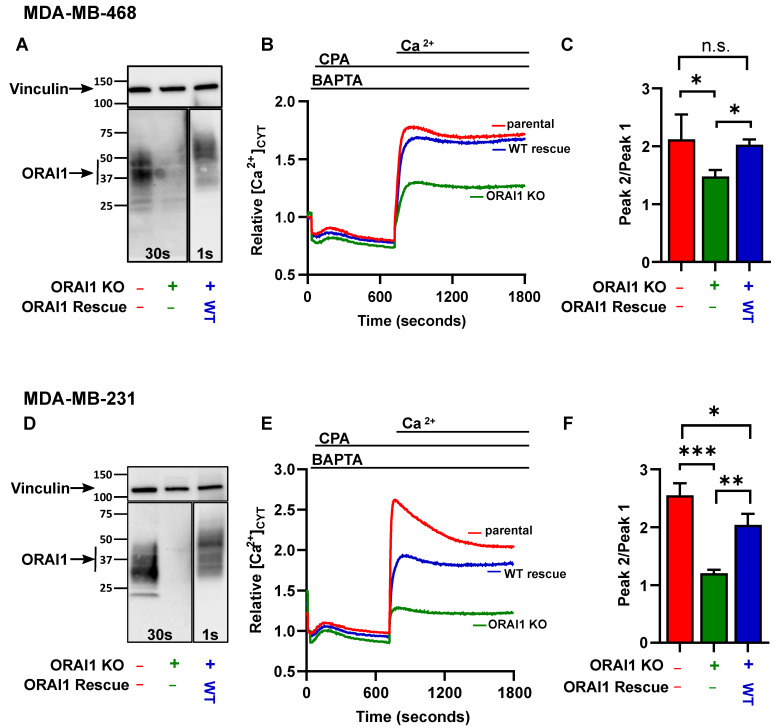
Generation of ORAI1 knockout and ORAI1 WT rescue breast cancer cell lines. (**A**) Representative immunoblot showing MDA-MB-468 endogenous ORAI1 expression, CRISPR-Cas9-mediated abolishment of ORAI1 expression, and rescue of ORAI1 expression in the ORAI1 knockout cells. (**B**) Representative traces of store-operated calcium entry (SOCE), represented as measurements of intracellular cytoplasmic Ca^2+^ ([Ca^2+^]_cyt_) in MDA-MB-468 parental cells (red), ORAI1 knockout cells (green), and ORAI1 WT rescue cells (blue). (**C**) Bar graphs showing peak2/peak1 (SOCE) from four independent experiments and representing mean ± SD (n.s., not significant (*p* ≥ 0.05); * *p* < 0.05, one-way ANOVA with Dunnett’s post-hoc). (**D**) Representative immunoblot showing ORAI1 expression in MDA-MB-231 cells. (**E**) Representative traces of SOCE in MDA-MB-231 parental cells (red), ORAI1 knockout cells (green), and ORAI1 WT rescue cells (blue). (**F**) Bar graphs showing peak2/peak1 (SOCE) from three independent experiments and representing mean ± SD (n.s., not significant (*p* ≥ 0.05); * *p* < 0.05; ** *p* < 0.05; *** *p* < 0.001, one-way ANOVA with Dunnett’s post-hoc).

**Figure 2 ijms-23-05867-f002:**
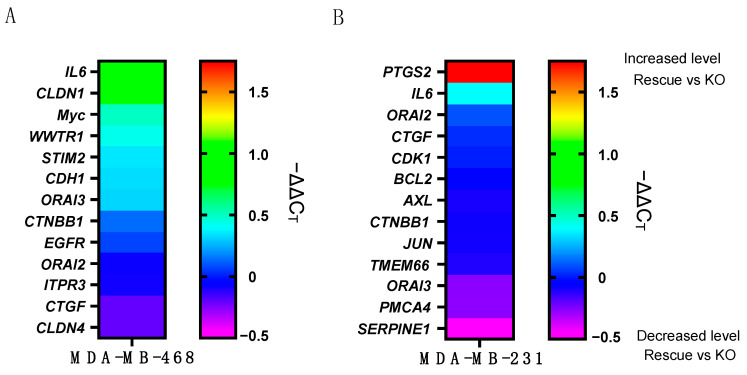
Effect of ORAI1 on gene expression. (**A**) Heatmap representing mRNA −ΔΔC_T_ of MDA-MB-468 ORAI1 WT rescue compared to MDA-MB-468 ORAI1 knockout cells. Data represent the mean of three independent experiments. (**B**) Heatmap representing mRNA −ΔΔC_T_ of MDA-MB-231 ORAI1 WT rescue compared to MDA-MB-231 ORAI1 knockout cells. Data represent the mean of three independent experiments.

**Figure 3 ijms-23-05867-f003:**
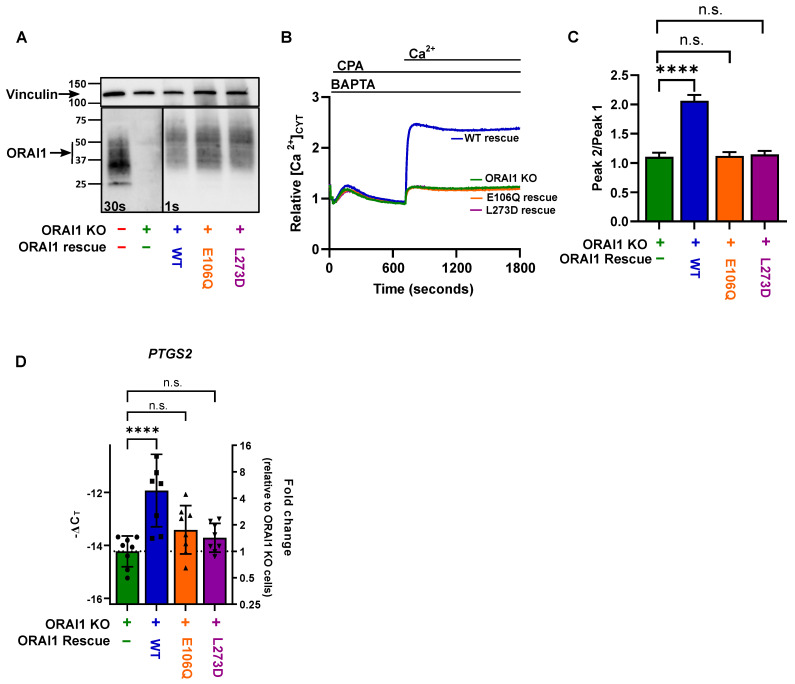
ORAI1 regulates *PTGS2* expression in MDA-MB-231 cells. (**A**) Representative immunoblot showing endogenous ORAI1 expression in MDA-MB-231 parental cells; ORAI1 knockout cells; and ORAI1 WT, ORAI1 E106Q, and ORAI1 L273D overexpression in MDA-MB-231 ORAI1 knockout cells. (**B**) Representative traces of store-operated calcium entry in MDA-MB-231 parental cells (red), ORAI1 knockout cells (green), ORAI1 WT (blue), ORAI1 E106Q (orange), and ORAI1 L273D (purple) rescue cells. (**C**) Bar graphs showing peak2/peak1 (SOCE) from three independent experiments and representing mean ± SD (n.s., not significant (*p* ≥ 0.05); **** *p* < 0.0001, one-way ANOVA with Dunnett’s post-hoc). (**D**) Bar graph showing −ΔC_T_ (left axis) and PTGS2 mRNA fold change (right axis) of eight independent experiments and representing mean ± SD (n.s., not significant (*p* ≥ 0.05); **** *p* < 0.0001, one-way ANOVA with Dunnett’s post-hoc performed on −ΔC_T_ values).

**Figure 4 ijms-23-05867-f004:**
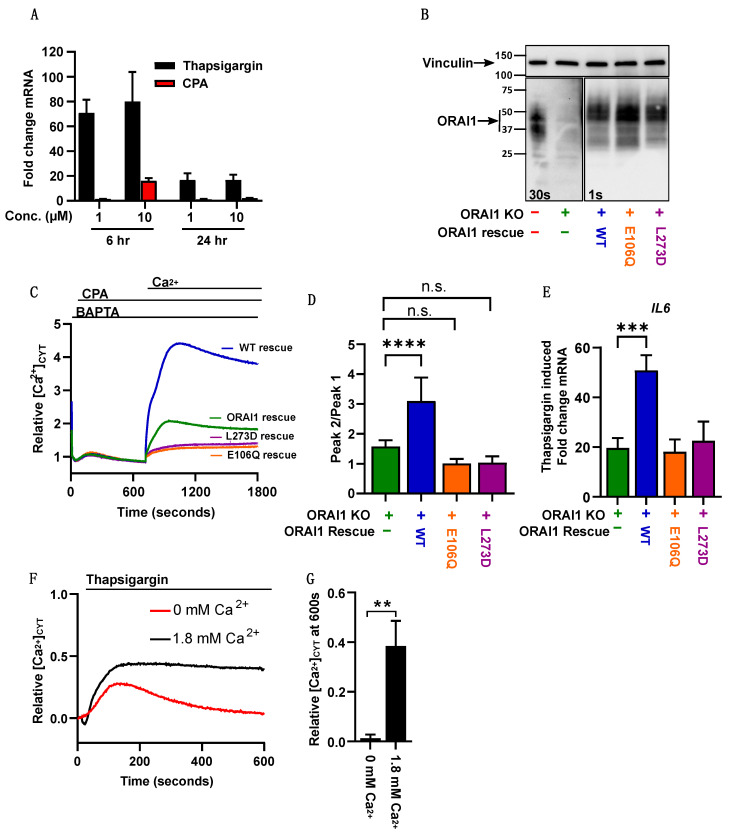
ORAI1 regulates *IL6* expression in MDA-MB-468 cells. (**A**) Bar graphs showing *IL6* mRNA level measured by RT-qPCR following treatment of MDA-MB-468 ORAI1 WT rescue cells with two concentrations of thapsigargin and CPA, at two different time points, from three independent experiments represented as mean ± SD. (**B**) Representative immunoblot showing endogenous ORAI1 expression in MDA-MB-468 parental cells; ORAI1 knockout cells; and rescue with ORAI1 WT, ORAI1 E106Q, and ORAI1 L273D overexpression in MDA-MB-468 ORAI1 knockout cells.(**C**) Representative traces of store-operated calcium entry in MDA-MB-468 parental cells (red), ORAI1 knockout cells (green), ORAI1 WT (blue), ORAI1 E106Q (orange), and ORAI1 L273D (purple). (**D**) Bar graphs showing peak2/peak1 (SOCE) from six independent experiments and representing mean ± SD (n.s., not significant (*p* ≥ 0.05); **** *p* < 0.0001, one-way ANOVA with Dunnett’s post-hoc). (**E**) Bar graphs showing *IL6* mRNA fold change induced by 6 h treatment with thapsigargin from five independent experiments and representing mean ± SD (*** *p* < 0.001, one-way ANOVA with Dunnett’s post-hoc). (**F**) Representative traces in MDA-MB-468 cells showing Ca^2+^ influx induced by thapsigargin in the absence (0 mM) or presence (1.8 mM) of extracellular Ca^2+^ from three independent experiments. (**G**) Bar graph showing the relative cytoplasmic calcium concentration at 600 s from three independent experiments and representing mean ± SD (** *p* < 0.001, *t*-test).

**Figure 5 ijms-23-05867-f005:**
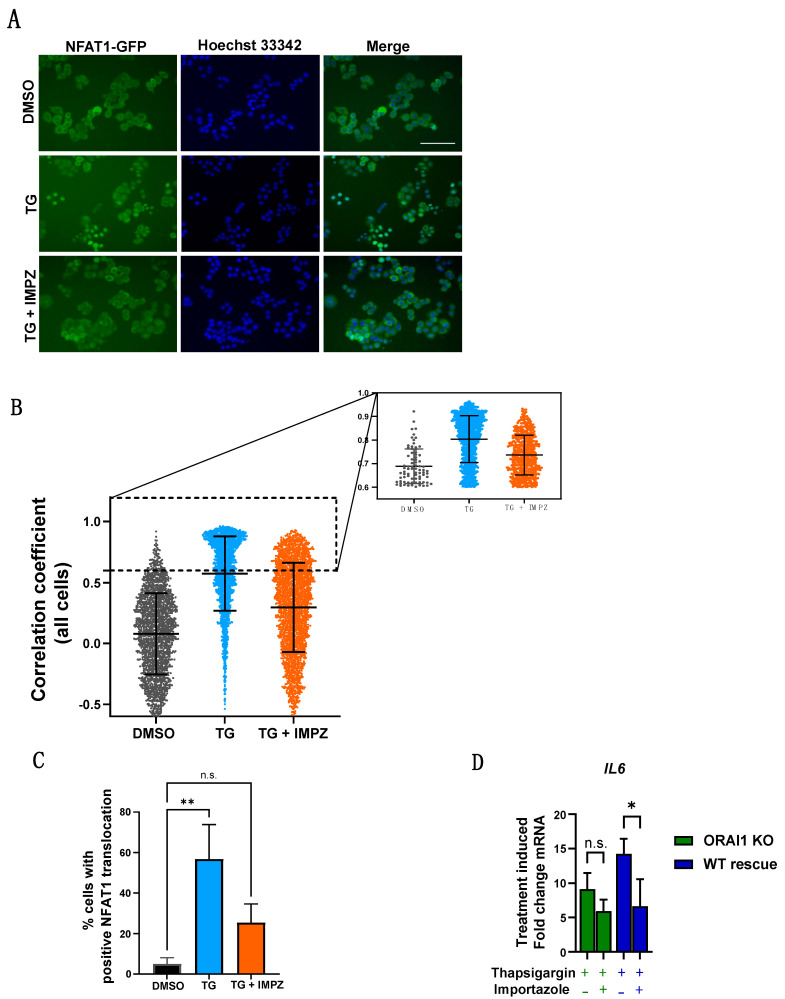
Importazole blocks thapsigargin-mediated NFAT1-GFP translocation. (**A**) Representative images of NFAT1-GFP (green) expressing MDA-MB-468 cells and Hoechst nuclear staining (blue) treated with DMSO control (top) and thapsigargin (TG) (middle) as well as pre-treated with importazole (IMPZ) for 15 min prior to 1 h TG treatment (bottom). Scale bar = 100 µm. (**B**) Quantification of single cell NFAT1-GFP translocation following treatment with DMSO, TG, or TG + IMPZ from three independent experiments and represents mean ± SD. Inset corresponds to cells with NFAT1-GFP positive translocation (correlation coefficient > 0.6). (**C**) Percentage of cells with NFAT1-GFP translocation from three independent experiments and represents mean ± SD (n.s., not significant (*p* ≥ 0.05); ** *p* < 0.01), one-way ANOVA with Dunnett’s post-hoc). (**D**) Bar graphs showing *IL6* mRNA fold change induced by 1 h treatment with thapsigargin or thapsigargin/importazole in MDA-MB-468 ORAI1 knockout and ORAI1 WT rescue cells from three independent experiments and representing mean ± SD (n.s., not significant, (*p* ≥ 0.05), * *p* < 0.05, two-way ANOVA with Šidák’s post-hoc).

## Data Availability

The data that support the findings of this study are available upon reasonable request from the corresponding author.

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
