# Peer review of "ORAI1-Regulated Gene Expression in Breast Cancer Cells: Roles for STIM1 Binding, Calcium Influx and Transcription Factor Translocation"

_ijms, 2022, doi:10.3390/ijms23115867_

Round 1

Reviewer 1 Report

This paper from Melanie Robitaille and colleagues describes the molecular role of ORAI1 in breast cancer cells. The authors first generated two orai1 knock out cell lines using Crisper-Cas9 strategy.   They found globe changes of key genes expression in orai1 KO cells compared to the wild type cell lines. Using ORAI1 mutants, the authors further found that pore function and interaction with STIM1 is important for ORAI1 mediated PTGS and IL6 expression in breast cell lines. Finally, they found ORAI1 is required for translocation of NFAT1 into nuclear.

ORAI1 was upregulated in many cancer cell lines including the ones used in this study. Therefore this study provides insights about how the molecular network changes in response to calcium intake mediated by ORAI1 channel.  Overall, the topic is in-line with the scope of the Journal. The paper is clear-structured and well-written. I think this is an interesting paper and only have a few comments to improve it prior to publication.

  1. I’m confused by two cell lines were chosen in this study? Further details are required to explain about why to choose them, and what’s the difference?
  2. All the main figures have very poor resolution, need significant improvement.
  3. In Figure 2: I’m a bit confused by different genes were plotted in A and B. Were different genes expressed in two cell lines or some genes were filtered out? If the latter is the case, please plot all the genes you have tested in these two cell lines
  4. Figure 2A: change the color for Thapsigargain and/or CPA treatment, as the original colors were too closed to each other.

Author Response

Thank you for the opportunity to allow us to improve our manuscript “ORAI1 regulated gene expression in breast cancer cells: role for STIM1 binding, calcium influx and transcription factor translocation”. We thank you for your valuable comments and have now addressed the comments as noted.

I’m confused by two cell lines were chosen in this study? Further details are required to explain about why to choose them, and what’s the difference?

We have added/modified the following sentences in section 2.1 from line 73, describing the rational and major morphological difference between the two cell lines: “As elevated ORAI1 expression is a feature of basal breast cancer cells3, we used basal A MDA-MB-468 and basal B MDA-MB-231 breast cancer cells19 in these studies. These are two widely used basal breast cancer cell lines. Basal A cell lines (eg.MDA-MB-468) display epithelial characteristics while basal B (eg.MDA-MB-231) display mesenchymal and stem/progenitor-like characteristics20”.

All the main figures have very poor resolution, need significant improvement.

Thank you for the valuable comment. We have improved the resolution of the figures.

In Figure 2: I’m a bit confused by different genes were plotted in A and B. Were different genes expressed in two cell lines or some genes were filtered out? If the latter is the case, please plot all the genes you have tested in these two cell lines.

We have added as a supplementary figure our full heatmap of genes assessed in the MDA-MB-468 cell line. A reference to supplementary figure 1 is now found on line 104 in the manuscript.

Figure 2A: change the color for Thapsigargain and/or CPA treatment, as the original colors were too closed to each other.

We believe the reviewer is referring to Figure 4 rather than Figure 2. We have changed the colours of Figure 4A, F and G as suggested (light grey to red).

Reviewer 2 Report

The manuscript describes identification of genes regulated by ORAI1 in basal breast cancer cells, their dependence on Ca2+ influx and STIM1 binding.

The study is well designed and conducted; I have no objections to the methodology and statistical analysis.

Did authors perform MTT on cell lines? Or other test of cell viability, proliferation and cytotoxicity?

Minor issue: Please specify when the text is concerned around protein and where it describes gene expression/function. Gene names should be therefore italicized. Please be consistent through all manuscript.

Minor spell check is advised, some double-spaces and typos in lines: 208, 236, 359, 368, etc

Author Response

Thank you for the opportunity to allow us to improve our manuscript “ ORAI1 regulated gene expression in breast cancer cells: role for STIM1 binding, calcium influx and transcription factor translocation”. We thank you for your valuable comments and have now addressed specific comments.

Did authors perform MTT on cell lines? Or other test of cell viability, proliferation and cytotoxicity?

We haven’t assessed the effect of ORAI1 knockout on cell viability. While this would be interesting, the focus of this paper is gene expression, and is was not feasible to complete MTT assays within the 5 days deadline provided by the journal.

Minor issue: Please specify when the text is concerned around protein and where it describes gene expression/function. Gene names should be therefore italicized. Please be consistent through all manuscript.

We have now addressed this comment, by italicizing gene names in the main text.

Minor spell check is advised, some double-spaces and typos in lines: 208, 236, 359, 368, etc

We have carefully reviewed the manuscript and corrected the double-spaces and typographical errors